# Airy Transform of the New Power-Exponent-Phase Vortex Beam

**Qidong Lin [1], Hao Zhang [1], Zhiquan Hu [1], Xiaotan Lu [1], Xingyuan Lu [1,*], Yangjian Cai [2,3] and Chengliang Zhao [1,*]**

[1] School of Physical Science and Technology, Soochow University, Suzhou 215006, China; 2022401071@stu.suda.edu.cn (Q.L.); 20204008017@stu.suda.edu.cn (H.Z.); 20224208022@stu.suda.edu.cn (Z.H.); 2022401057@stu.suda.edu.cn (X.L.)

[2] Shandong Provincial Engineering and Technical Center of Light Manipulations & Shandong Provincial Key Laboratory of Optics and Photonic Device, School of Physics and Electronics, Shandong Normal University, Jinan 250358, China; yangjiancai@sdnu.edu.cn

[3] Joint Research Center of Light Manipulation Science and Photonic Integrated Chip, East China Normal University, Shanghai 200241, China

\* Correspondence: xylu@suda.edu.cn (X.L.); zhaochengliang@suda.edu.cn (C.Z.)

**Abstract:** A new power-exponent-phase vortex beam with nonlinear phase winding has shown flexible control freedom compared with conventional vortex beams. In order to further enrich the modulation freedom and expand the ability of self-healing to meet current application requirements, we conducted a detailed study on the characteristics of the Airy transform of the new power-exponent-phase vortex beam. The influences of the Airy function, the power exponent, and the topological charge on normalized intensity and phase distributions are investigated theoretically and experimentally. More importantly, the self-healing properties of the new power-exponent-phase vortex beam with and without the Airy transform are compared. This shows that the new power-exponent-phase vortex beam with the Airy transform exhibits better self-healing ability when obstructed by obstacles. This study has potential applications in optical trapping and free-space optical communication.

**Keywords:** airy transform; new power-exponent phase; vortex beam; self-healing





## 1. Introduction

The optical vortex was first proposed by Coullet in 1989 [1]. An optical vortex beam carries a phase taking the form of $e^{im\varphi}$, where $m$ represents the topological charge (TC), which refers to the number of twists in a wavefront per unit of wavelength, and $\varphi$ is the azimuthal coordinate in the transverse plane. Its wavefront is helical, while the intensity distribution is ring-shaped [2], and during transmission, the beam has a phase singularity in the center, wherein the light intensity is zero [3]. Compared with ordinary beams, vortex beams have aroused strong interest from researchers since their origin because of their distinctive characteristics and have quickly become an important branch of modern optical research. In 1992, Allen demonstrated that each photon in a vortex beam carries an orbital angular momentum (OAM) of $m\hbar$, where $\hbar$ is the reduced Planck constant [4]. The OAM can be transmitted to the particles, and then the particles are trapped and driven to rotate [5]. In addition, OAM corresponds to an infinite number of eigenvalues, and it is theoretically possible to construct an infinite-dimension Hilbert space using vortex beams with various TCs [6–9], which greatly improves the capacity and spectrum utilization of communication systems. In addition, the uncertain relationship between TC and azimuth makes the communication confidential [6]. With its unique optical properties, the vortex beam is currently used in the fields of optical trapping [10–12], optical communication [13,14], optical detection [15], optical imaging [16,17], and quantum entanglement [18–20].

Over the past decades, apart from the conventional optical vortex beam, several kinds of noncanonical optical vortices, such as Mathieu vortex [21], fractional vortex [22], Bessel

vortex [23], power-exponent-phase vortex [24], partially coherent vortex [25–27], and so on [28], have been researched to explore the properties and applications of OAM. A new power-exponent-phase vortex (NPEPV) beam, as one of the new models for vortex beams, has shown rich modulation dimensions and realized diverse intensity patterns, such as polygonal and porous distributions [29]. The spatial light field distribution of the NPEPV beam is determined not only by the value of the TC but also by the power exponent.

In practical optical communication applications, it is common for a beam to encounter disturbation during propagation. In this case, the NPEPV beam will lose some information, thus greatly reducing the accuracy of information transmission [30,31]. Airy beams have the characteristics of nondiffraction, self-healing, and self-acceleration, making them ideal light sources in the case above [32,33]. Several methods exist for generating Airy beams, the most basic of which are based on the spatial light modulator (SLM). Among the generation methods based on the SLM, the Airy transform is the simplest one [34]. The Airy transform of a beam can endow the initial beam with Airy characteristics. For example, an Airy beam can be obtained by the Airy transform of the most common Gaussian beam [34]. The special properties of the Airy beam have attracted enormous research interest, and some researchers have imparted Airy properties to the vortex beam, such as Gaussian vortex beam, Laguerre–Gaussian beam, Hermite–Gaussian beam, and Lorentz–Gauss beam [24,35–37], resulting in the Airy vortex beam.

Owing to the self-healing properties of the Airy beam, we performed an Airy transform on the NPEPV beam. A detailed study on the characteristics of the Airy transform of the NPEPV beam was conducted theoretically and experimentally. The effects of the Airy control parameter, the power exponent and topological charge on the normalized intensity distribution, and the phase distribution of the Airy transform of the NPEPV beam were investigated, especially for the centroid shift and beam spot size. In addition, the self-healing properties of the NPEPV beam with and without the Airy transform were compared. The results show that the Airy transform has a beneficial impact on the self-healing properties of NPEPV beams.

## 2. Theory

The electric field of an NPEPV beam in the input plane is described by [29]:

$$E_{m,n}(x_0, y_0) = \exp\left(-\frac{x_0^2 + y_0^2}{w_0^2}\right) \exp\left\{i2\pi\left[\frac{\mathrm{rem}(m\varphi, 2\pi)}{2\pi}\right]^n\right\} \tag{1}$$

where $x_0$ and $y_0$ are coordinates in the input plane, $w_0$ means the Gaussian waist, rem(.) denotes the remainder function, $n$ is the power exponent, $m$ is the value of the TC, and $\varphi$ is the azimuthal coordinate in the transverse plane. The phase and intensity patterns of conventional vortex beam ($m = 2$ and $n = 1$, Figure 1(a1,a2)) and NPEPV beam ($m = 2$ and mboxemph$n = 2$, Figure 1(b1,b2)) are compared.

The electric field characterizing the NPEPV beam's propagation through the Airy transform optical system can be described by [34]

$$E_{m,n}(x, y) = \frac{1}{|\alpha\beta|} \iint\limits_{-\infty}^{+\infty} E_{m,n}(x_0, y_0) Ai\left(\frac{x - x_0}{\alpha}\right) Ai\left(\frac{y - y_0}{\beta}\right) dx_0 dy_0, \tag{2}$$

where $\alpha$ and $\beta$ are the Airy control parameters in the $x$- and $y$-directions, respectively, $x$ and $y$ are the coordinates in the output plane. $Ai(.)$ denotes Airy function defined by

$$Ai(x) = \frac{1}{\pi}\int_0^\infty \cos\left(\frac{t^3}{3} + xt\right) dt. \tag{3}$$

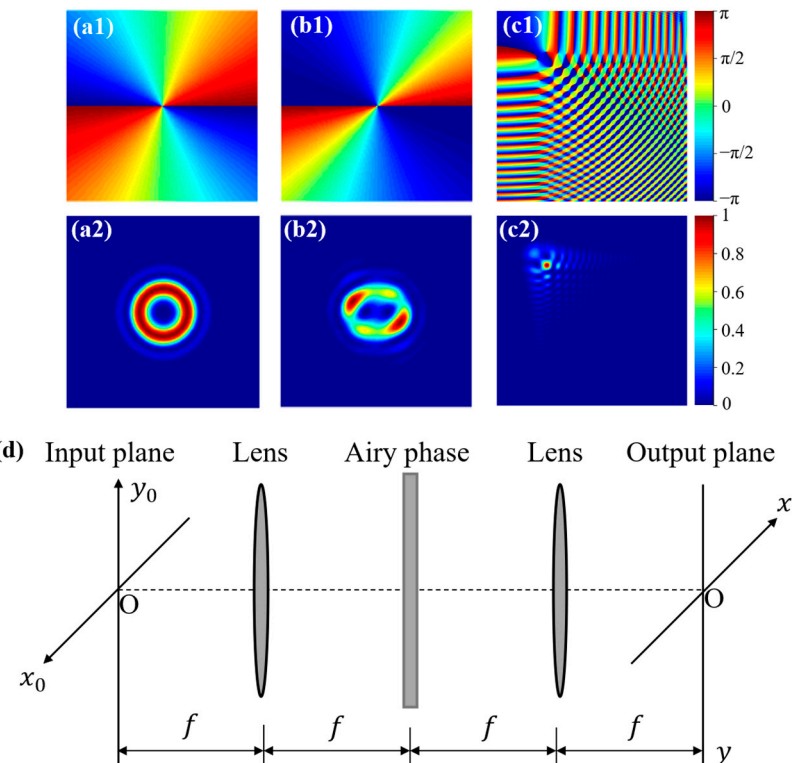

**Figure 1.** Schematic diagram of the Airy transform optical system. Intensity and phase patterns of (**a1**,**a2**) the conventional vortex beam ($m$ = 2 and $n$ = 1), (**b1**,**b2**) the new power-exponent-phase vortex beam ($m$ = 2 and $n$ = 2), and (**c1**,**c2**) the new power-exponent-phase vortex beam with Airy transform ($m$ = 2, $n$ = 2 and $\alpha = \beta = 0.3$ mm). (**d**) Airy transform optical system.

Typical intensity and phase patterns are shown in Figure 1(c1,c2). Figure 1d depicts an example of an optical system employed to achieve the Airy transform [34]. This optical arrangement comprises a pair of thin lenses with a focal length $f$, forming a 4f optical system. Additionally, a spatial light modulator in the back focal plane of the first lens could be used to introduce the Airy transform phase, which can be calculated by Fourier transform of $Ai\left(\frac{x-x_0}{\alpha}\right)Ai\left(\frac{y-y_0}{\beta}\right)$.

Then, the intensity and phase of the Airy transform of the NPEPV beam can be obtained, respectively, with $I_{m,n}(x,y) = E^*_{m,n}(x,y)E_{m,n}(x,y)$, $P_{m,n}(x,y) = Arg[E_{m,n}(x,y)]$. The centroids in two transverse directions of the beam in the output plane are given by [38]:

$$X_{m,n} = \frac{\int_{-\infty}^{\infty}\int_{-\infty}^{\infty} x \cdot I_{m,n}(x,y)dxdy}{\int_{-\infty}^{\infty}\int_{-\infty}^{\infty} I_{m,n}(x,y)dxdy}, \quad Y_{m,n} = \frac{\int_{-\infty}^{\infty}\int_{-\infty}^{\infty} y \cdot I_{m,n}(x,y)dxdy}{\int_{-\infty}^{\infty}\int_{-\infty}^{\infty} I_{m,n}(x,y)dxdy}. \tag{4}$$

The beam spot sizes in the two transverse directions are defined by [38]:

$$W_{m,n,x} = \left[\frac{\int_{-\infty}^{\infty}\int_{-\infty}^{\infty} x^2 I_{m,n}(x,y)dxdy}{\int_{-\infty}^{\infty}\int_{-\infty}^{\infty} I_{m,n}(x,y)dxdy} - X^2_{m,n}\right]^{1/2},$$

$$W_{m,n,y} = \left[\frac{\int_{-\infty}^{\infty}\int_{-\infty}^{\infty} y^2 I_{m,n}(x,y)dxdy}{\int_{-\infty}^{\infty}\int_{-\infty}^{\infty} I_{m,n}(x,y)dxdy} - Y^2_{m,n}\right]^{1/2}. \tag{5}$$

To investigate the self-healing characteristics of the Airy beam, we considered a scenario where the initial beam is partially obstructed by a finite opaque obstacle [33,39].

The obstacle was assumed to have a square profile. Consequently, the transmittance function of the output beam can be described as follows:

$$M(x,y) = \begin{cases} 0, & |x - b_x| + |y - b_y| < d/2 \\ 1, & |x - b_x| + |y - b_y| \geq d/2 \end{cases}, \tag{6}$$

where $d$ is the size and $(b_x, b_y)$ are the central coordinates of the opaque obstacle. Using the Fresnel diffraction integral, the electric field on the z plane can be expressed by [33]:

$$E_{m,n}(x_1, y_1) = \frac{k}{2\pi i z} \int_{-\infty}^{\infty} \int_{-\infty}^{\infty} E_{m,n}(x,y) M(x,y) e^{\{\frac{ik}{2z}[(x_1 - x)^2 + (y_1 - y)^2]\}} dx dy, \tag{7}$$

where $x_1$ and $y_1$ are two transverse coordinates on the z plane, $k = 2\pi/\lambda$, and $\lambda$ is the wavelength.

## 3. Simulation Results

The characteristics of the Airy transform of the NPEPV beams with different TCs, power exponent, and Airy control parameters were analyzed using numerical examples based on the formulas in the previous section. For simplicity, we considered a fixed Gaussian waist of $w_0 = 0.5$ mm in the following analysis. We assumed that both $\alpha$ and $\beta$ are positive, and we focused on the symmetric case of $\alpha = \beta$.

The effects of the Airy control parameters on the normalized intensity distribution and phase distribution of the NPEPV beam with $m = 1$ and $n = 2$ passing through the Airy-transformed optical systems were investigated, as shown in Figure 2. Figure 2(a1–d1) show that when the values of $\alpha$ and $\beta$ are 0.5 mm, the beam spot exhibits characteristic Airy patterns. As $\alpha$ and $\beta$ decrease, the Airy patterns progressively diminish until they are completely absent, giving way to the emergence of a dominant NPEPV pattern. The corresponding phase distributions are shown in Figure 2(a2–d2). The phase singularity is located at the point of zero intensity (marked in white arrows) in the main lobe, and its winding number corresponds to the absolute value of TC.

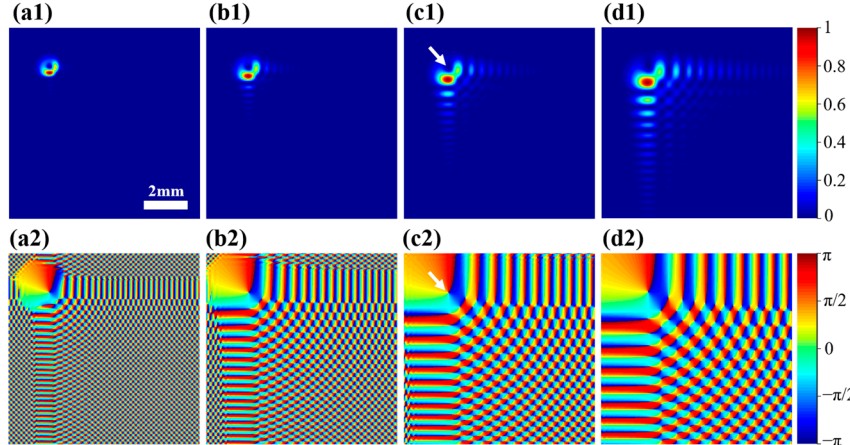

**Figure 2.** Normalized intensity distribution and phase distribution of the NPEPV beam with $m = 1$ and $n = 2$ passing through the Airy-transformed optical systems with different sizes of $\alpha$ and $\beta$. (**a1,a2**) $\alpha = \beta = 0.2$ mm; (**b1,b2**) $\alpha = \beta = 0.3$ mm; (**c1,c2**) $\alpha = \beta = 0.4$ mm; and (**d1,d2**) $\alpha = \beta = 0.5$ mm. White arrows mark the intensity and phase singularity.

Figure 3 depicts the normalized intensity and phase distributions in the output plane for different TCs ($m = 1, 2$), power exponent values ($n = 2, 5, 8$), and Airy control parameters ($\alpha = \beta = 0.3$ mm, 0.5 mm). The remaining parameters are the same as Figure 2. The intensity and phase distribution of the output beam are identical when the absolute values of TC are the same. As shown in Figure 3(d1,d2,j1,j2), when $m = 2$ and $n = 2$, the main lobe of the image deviates from the center, and the phase singularity splits into two

singularities (see zoom-in figure in Figure 3(d1,d2)). As the absolute value of TC increases, the Airy feature strengthens, and the beam spot size increases. Meanwhile, the details and hierarchies of each lobe in the phase distribution become more intricate. As *n* increases, the main lobe of the beam converges toward the center of the image, forming a Gaussian-like shape, which is a manifestation of the characteristic of the NPEPV. Meanwhile, the energy proportion of the side lobes gradually decreases.

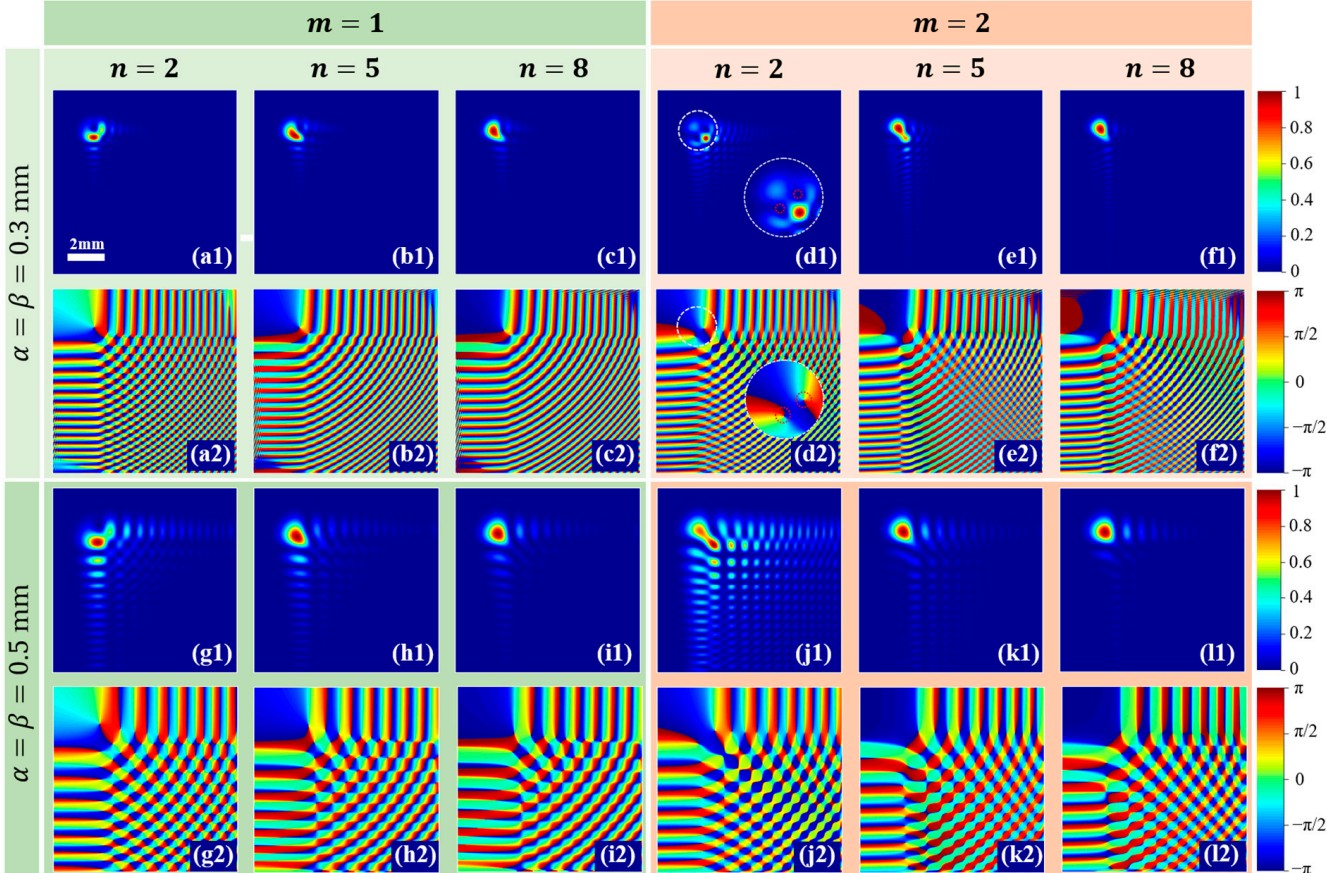

**Figure 3.** Normalized intensity distribution and phase distribution of the NPEPV beam with different TCs, *m*, and power exponent parameters *n* after passing through the Airy-transformed optical systems with (**a1–f1,a2–f2**) $\alpha = \beta = 0.3$ mm and (**g1–l1,g2–l2**) $\alpha = \beta = 0.5$ mm.

In Figure 4, we quantitatively examined the impact of the Airy control parameters and the TC on the centroid and beam spot size of the NPEPV beams with $n = 2$, $m = \pm 1, \pm 2$, and $\pm 3$ after the Airy transform with $\alpha = \beta = 0.5$ mm. In this analysis, we focused on the *x*-direction only because the centroid and beam spot size exhibit similar variations in both transverse directions. As depicted in Figure 4, the sign of the centroid in the *x*-direction is the same as that of the Airy control parameter $\alpha$. As the absolute value of $\alpha$ increases, the centroid deviates further from the origin. The beam spot size in the *y*-direction is independent of the sign of $\alpha$. When the absolute value of $\alpha$ decreases, the beam spot size in the *x*-direction gradually decreases. In addition, as the absolute value of TC *m* increases, both the centroid and beam spot size in the *x*-direction increase.

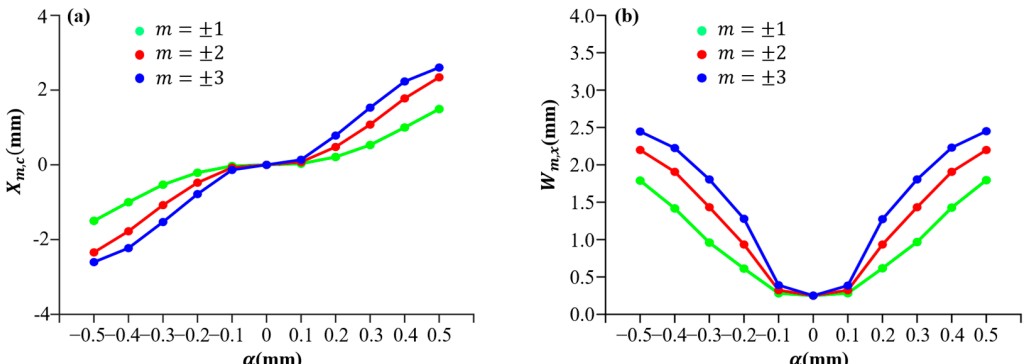

**Figure 4.** Theoretical results of the (**a**) centroid and (**b**) beam spot size in the *x*-direction of the NPEPV beams after the Airy transform as a function of the Airy control parameter $\alpha$.

Considering the self-healing capability of the Airy beam, we compared the self-healing properties of the NPEPV beam with and without the Airy transform. $\alpha = \beta = 0.5$ mm, $n = 2$, and $m = 1$. The opaque obstacle was placed 0.3 m behind the output plane and blocked 33% of the energy of the light field, as shown with the red box in Figure 5. Figure 5a,c show the main lobe images of the intensities after further propagation to 0.5, 1, 1.5, and 2 m. Figure 5b,d show the corresponding lobe images after being blocked by the opaque occlusion. By comparing the intensities at the same distance, it can be observed that after applying the Airy transform, the NPEPV beam exhibits better self-healing effects compared to that without the Airy transform. To quantitatively demonstrate this characteristic, Figure 5e displays the similarity curves obtained by calculating the similarity of the beams before or after the obstruction at different propagation distances using the built-in similarity calculation function (SSIM function in MATLAB). The similarity curves align well with the qualitative observations.

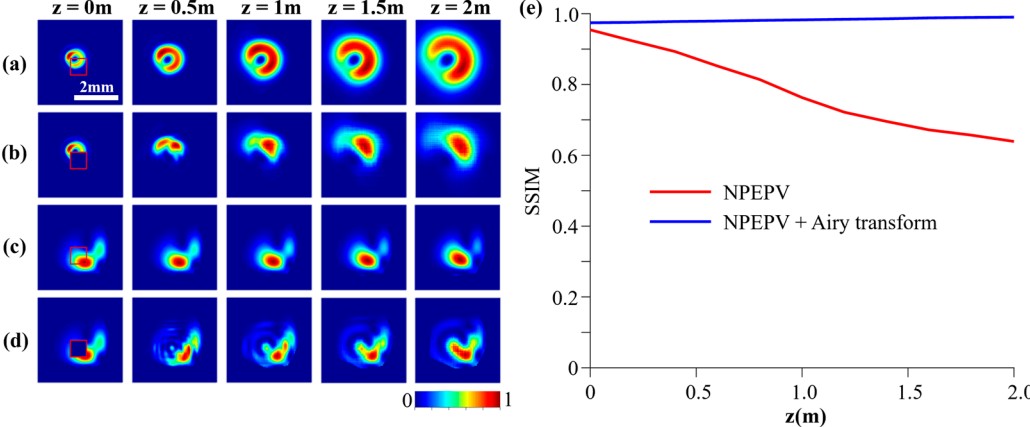

**Figure 5.** Self-healing characteristics of the NPEPV beam with and without the Airy transform. The main lobe images of the NPEPV beam (**a**) without and (**b**) with an opaque obstacle and the Airy transform of the NPEPV beam (**c**) without and (**d**) with an opaque obstacle. The graph (**e**) illustrates the similarity of the main lobe images with and without the obstacle at various propagation distances. Red, the NPEPV beam and blue, the Airy transform of the NPEPV beam. $\alpha = \beta = 0.5$ mm, $n = 2$, and $m = 1$. The obstacles block 33% of the total energy.

## 4. Experimental Results

In this section, we conducted an experiment to clarify the characteristics of the Airy transform of the NPEPV beam. The experiment setup for the generation of the NPEPV beam, achievement of the Airy transform, and mensuration of the output beam are illustrated in Figure 6. A linear Gaussian beam emitted from a He–Ne laser with the wavelength of 532 nm is first expanded through a beam expander (BE), and after being reflected by the reflection mirror (RM), the beam directs toward a spatial light modulator (SLM1) controlled

by a computer. The computer is used to load the hologram of the NPEPV beam [40]. The scatting light after SLM1 passes through a 4f optical system composed of lenses L1 and L2. An aperture is placed on the focal plane of L1, allowing only first-order diffraction to pass through. In the focal plane of L2, it can be regarded as an NPEPV beam, whose waist width, TC, and power exponent are specified by the computer hologram on SLM1.

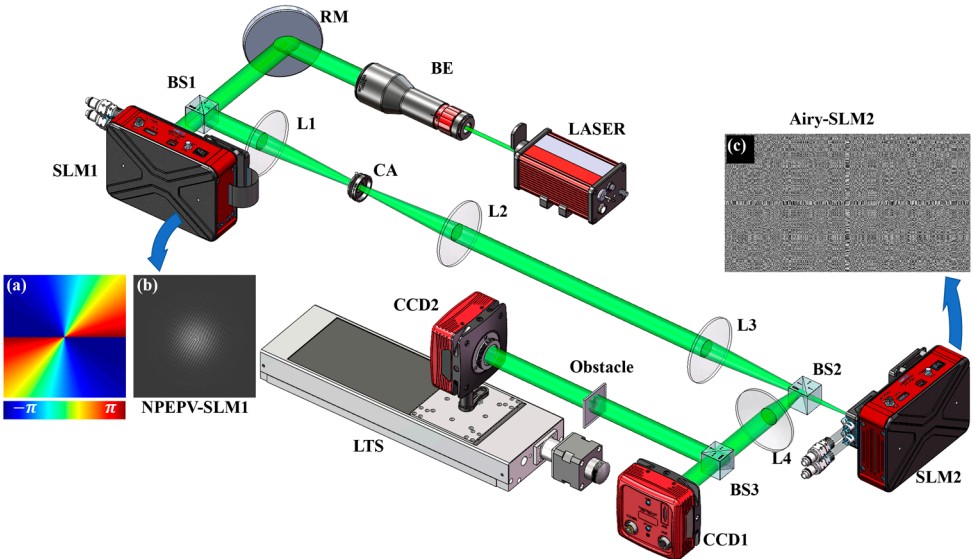

**Figure 6.** Experimental setup for the Airy transform of the NPEPV beam. (**a**) Phase of the NPEPV beam with TC = 2 and $n$ = 2; (**b**) Hologram loaded on SLM1; (**c**) Airy transform phase loaded on SLM2. LASER, coherent light source with wavelength of 532 nm; BE, beam expander; RM, reflective mirror; BS1, BS2, and BS3, beam splitters; L1, L2, L3, and L4, lenses; SLM1 and SLM2, spatial light modulator; CCD1, charge-coupled device used for intensity monitoring on the output plane; CCD2, charge-coupled device for intensity capture with obstacles; LTS, linear translation stage.

Next, the NPEPV beam passes through the Airy transform optical system consisting of lenses L3 and L4. SLM2 is loaded with a hologram calculated with the Fourier transform of the Airy phase, i.e., $\psi(x_1, y_1) = \left(\alpha^3 k^3 x_1^3 + \beta^3 k^3 y_1^3\right)/\left(3f_3^3\right) - (2kf_3 + 2kf_4 + \pi)$, where $f_3 = 400$ mm and $f_4 = 400$ mm are the focal lengths of L3 and L4, respectively. In the focal plane of L4, the output beam, whose intensity distribution is captured by CCD1, can be regarded as an NPEPV beam with an Airy transform. An opaque obstacle was placed 0.3 m away from the focal plane of L4, and CCD2 was used to capture the intensity distribution of the beam at different positions.

Figure 7 shows the experimental results of output intensities for different topological charges ($m$ = 1, 2) and various power exponent parameters ($n$ = 2, 5, 8) with $\alpha = \beta = 0.3$ mm and $\alpha = \beta = 0.5$ mm. The distribution and evolution characteristics fit well with theoretical simulations shown in Figure 3. The energy of the side lobe gradually weakens with the rising $n$, and this energy converges to the main lobe, causing the output beam to approach the Airy-like beam. As shown in Figure 7(a1,a2,d1,d2), when $m$ increases, the main lobe and the outermost side lobes of the output beam undergo varying degrees of splitting with singularity splitting.

Figure 8 shows the experimental self-healing results of the NPEPV beam with and without the Airy transform. $\alpha = \beta = 0.5$ mm, $n = 2$, and $m = 1$. The obstacles used to block the main lobe are shown with a red box in Figure 8 ($z = 0$). Then, we compared the self-healing ability of the main lobe of the NPEPV beam with and without the Airy transform. Figure 8a,b show the intensities of the NPEPV beam with and without obstacles at the same distance. Evidently, the pattern permanently loses a portion of its spot after being obstructed. However, in Figure 8d, as seen in the self-healing process of the NPEPV beam with the Airy transform, after the left half of the main lobe is obscured, the remaining spot

gradually "shifts" toward the obscured area and finally aligns with the freely transmitted spot in Figure 8c.

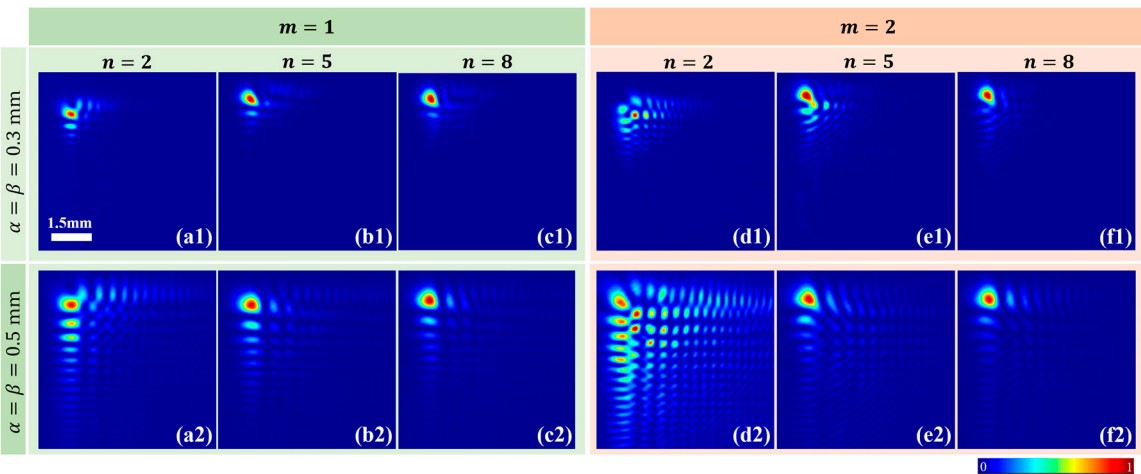

**Figure 7.** Experimental result of normalized intensity distribution and phase distribution of the NPEPV beam passing through the Airy-transformed optical systems. (**a1–c1**) $m = 1$ and $\alpha = \beta = 0.3$ mm; (**a2–c2**) $m = 1$ and $\alpha = \beta = 0.5$ mm; (**d1–f1**) $m = 2$ and $\alpha = \beta = 0.3$ mm; (**d2–f2**) $m = 2$ and $\alpha = \beta = 0.5$ mm.

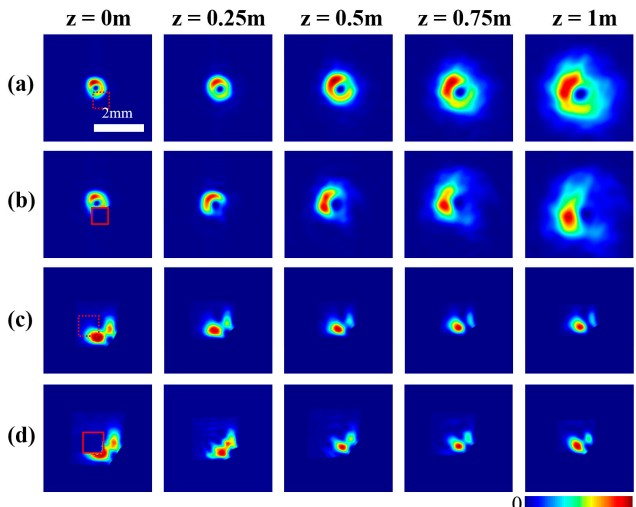

**Figure 8.** Experimental verification of the self-healing characteristics. The main lobe images of NPEPV beams (**a**) without and (**b**) with an opaque obstacle and the Airy transform of NPEPV beams (**c**) without and (**d**) with an opaque obstacle at different propagation distances. $\alpha = \beta = 0.5$ mm, $n = 2$ and $m = 1$.

## 5. Discussion

As illustrated in Equation (2), the values of $\alpha$ and $\beta$ control the Airy features in the horizontal and vertical directions, respectively. In the simulation and experiment, $\alpha$ and $\beta$ with the same values were considered. When their values are different, the Airy features will be asymmetric in the horizontal (or $x$) and vertical (or $y$) directions. In addition, for positive $\alpha$ (or $\beta$), sidelobes show in the $+x$ (or $+y$) direction, while for negative case, sidelobes will flip to the opposite direction. Regarding the self-healing properties of the NPEPV beam after the Airy transform, more obstructions were tested by simulation, as shown in Figure 9. Figure 9b,d correspond to a random-point obstruction, and Figure 9g,i correspond to a leaf-shape obstruction. Due to the Airy transform, the shape of the NPEPV beam is modulated compared with the conventional vortex beam. Therefore, for both types of obstructions, obstacles block 40% of the total energy, and in both cases, the NPEPV beam

shows better self-healing properties. However, it must be pointed out that self-healing here cannot achieve the reconstruction of the source field information, but only makes the transmission characteristics appear close to being unobstructed. The NPEPV beams with the Airy transform proposed in this article carry multidimensional information, including topological charges, power exponent parameters, and airy parameters of horizontal and vertical directions. The self-healing characteristics analyzed here have advantages in the applications of inferring or recognizing optical field information based on transmission characteristics. Moreover, compared with conventional vortex beams, the identification of topological charges after the Airy transformation is a problem worth further research [41,42]. Whether the topological charges can be identified visually depends on the domination of the Airy parameters or the NPEPV parameters. It is expected to be solved in optical systems based on learning-based recognition [43].

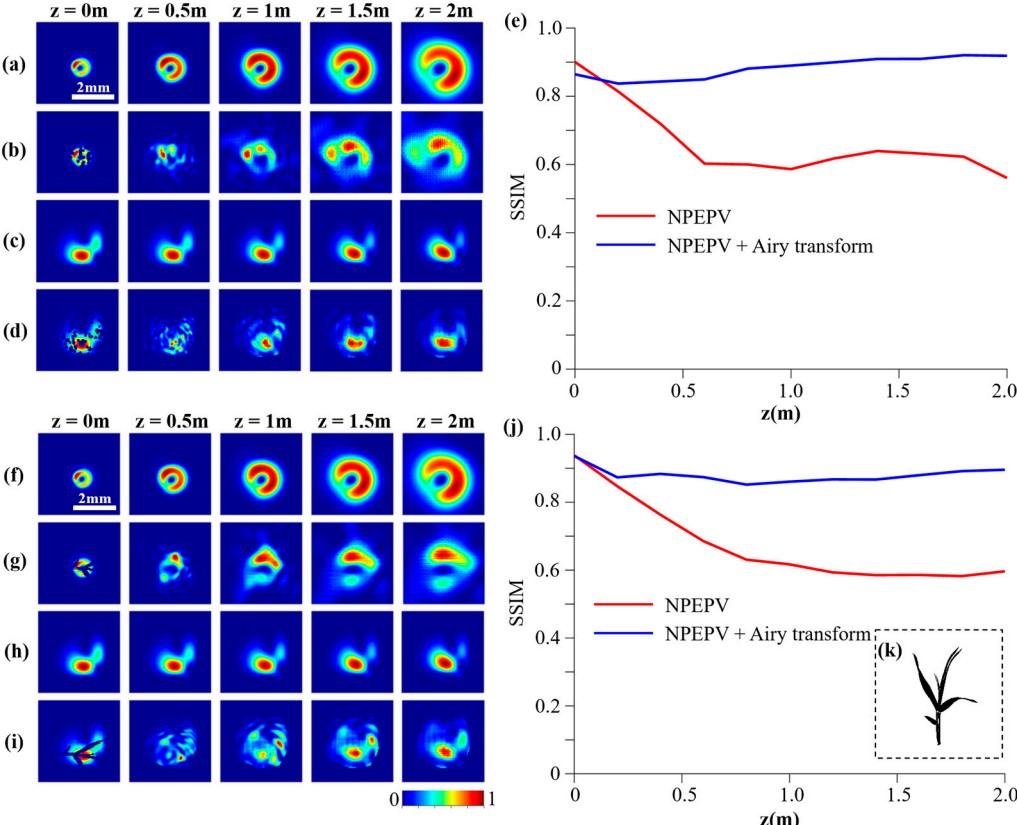

**Figure 9.** Self-healing characteristics of the NPEPV beam with and without the Airy transform for different obstructions. The main lobe images of the NPEPV beam (**a**,**f**) without and (**b**,**g**) with random-point obstacles and leaf-shape obstacles and the Airy transform of the NPEPV beam (**c**,**h**) without and (**d**,**i**) with random-point obstacles and leaf-shape obstacles. The graphs (**e**,**j**) illustrate the similarity of the main lobe images with and without the obstacles at various propagation distances. Red, the NPEPV beam and blue, the Airy transform of the NPEPV beam. (**k**) Leaf-shape obstacle. For all cases, obstacles block 40% of the total energy.

## 6. Conclusions

The Airy transform of the NPEPV beam is studied in this work. The influences of the Airy control parameters, the power exponent, and the TC on the normalized intensity and phase distribution are investigated theoretically. The quantification simulations show that the absolute value of the centroid shift and beam spot size of the NPEPV beam after the Airy transform increase with the Airy control parameters and TC. More importantly, the self-healing properties of the NPEPV beam with and without the Airy transform were studied theoretically and experimentally. The results show that the NPEPV beam after the Airy transform has a better self-healing ability than the case without the Airy

transform. Thus, the Airy transform of the NPEPV beam can have a beneficial impact on the research of vortex beams, which can further accelerate its applications in the field of optical communication.

**Author Contributions:** Conceptualization and methodology, Q.L., H.Z. and X.L. (Xingyuan Lu); writing—original draft preparation, Q.L. and X.L. (Xingyuan Lu); supervision, X.L. (Xingyuan Lu), C.Z. and Y.C.; funding acquisition, X.L. (Xingyuan Lu), C.Z. and Y.C. All authors contributed to the writing, review, and editing. All authors have read and agreed to the published version of the manuscript.

**Funding:** National Key Research and Development Program of China (No. 2022YFA1404800, No. 2019YFA0705000), National Natural Science Foundation of China (No. 12204340, No. 12174280, No. 12192254, No. 11974218, No. 92250304), China Postdoctoral Science Foundation (2022M722325), Priority Academic Program Development of Jiangsu Higher Education Institutions, Key Lab of Modern Optical Technologies of Jiangsu Province (KJS2138), and Undergraduate Training Program For Innovation and Entrepreneurship, Soochow University (202210285127Y).

**Institutional Review Board Statement:** Not applicable.

**Informed Consent Statement:** Not applicable.

**Data Availability Statement:** Data underlying the results presented in this paper are not publicly available at this time but may be obtained from the authors upon reasonable request.

**Conflicts of Interest:** The authors declare no conflict of interest.

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
