# Peer review of "Airy Transform of the New Power-Exponent-Phase Vortex Beam"

_photonics, doi:10.3390/photonics10090974_

Round 1
Reviewer 1 Report
In the work entitled “Airy transform of the new power-exponent-phase vortex beam”, the authors conducted a detailed study on the characteristics of the Airy transform of the new power-exponent-phase vortex beam. The influences of the Airy function, the power exponent and the topological charge on intensity and phase distributions are investigated theoretically and experimentally.
This work revealed the remarkable advantages of Array transform in self-healing properties of new power-exponent-phase vortex beam and proved it in experiment. It shows that the new power-exponent-phase vortex beam with Airy transform exhibits better self-healing ability when obstructed by obstacles. The results are elegant and highly convincing. The manuscript is well organized and clearly written. I have no reservations to recommend its publication in Photonics. Here are some minor issues:
1. Regarding the self-healing properties, have you tried different types of obstructions, different positions or different proportions of occlusion? I’ll suggest to add some discussions in the Conclusion part.
2. Compared with conventional vortex beams, new power-exponent-phase vortex beams have special phase winding. To avoid confusion, I’ll suggest the authors to add some phase patterns of the original the new power-exponent-phase vortex beam in Fig. 3.
Reviewer 2 Report
The Airy transform of a new power-exponent-phase vortex (NPEPV) beam is studied in this manuscript (MS). The effects of the Airy function, the power exponent and the topological charge of the NPEPV beam on the field distribution are discussed both theoretically and experimentally. It is shown that the self-healing ability of the NPEPV beam after Airy transform is improved. The finding would provide a way to control the vortex beam in optical communication.
The MS is written well and the analysis is quite clear. There may exist some issues that the author should consider for improving the quality of this MS.
1. The new power-exponent-phase vortex (NPEPV) beam adopted in this MS is not very well-known for most of readers, so it would be better to add some plots to show the main character of this beam before the Airy transform.
2. In the last paragraph of page 4, it was said that in Fig. 4, m = \pm 1, \pm 2 (positive and negative values), while when we looked at Fig. 4, there were only positive values of m. The author needs to explain or clarify this inconsistency.
Also, in this figure, \alpha was changed from -0.5 to +0.5. So, what is the physical meaning of the negative value of this parameter? Or why is it interesting to examine the negative value of \alpha?
3. In this MS, only the case that the \alpha and \beta hold the same value was considered. Do the main results or the rules obtained in this MS would be changed if they had different values?
Reviewer 3 Report
In this paper, the authors show through modeling and experimentally that the vortex beam recovers better after being distorted by an obstacle if it has undergone an Airy transformation. In this work, all formulas (1)-(7) are well known. Therefore, there is no theoretical novelty in this work. The only novelty is that the Airy transformation (2) improves the self-healing of the optical vortex (1). But why this happens the authors do not consider. It seems to me that the comparison of self-healing in figures 5 and 8 is not correct. Since after the Airy transformation the center of the beam shifts and the beam itself broadens (Fig.4), the obstacle cuts out a smaller fraction of the energy from the beam. Therefore, the reconstruction of the beam after the Airy transformation is better. This is clearly seen from Fig.5e. The blue curve in Fig. 5e is almost constant. This means that the obstacle almost did not distort the beam. A correct comparison of the beam reconstruction (1) without the Airy transform (2) and with the Airy transform will be if the obstacle cuts out the same part of the beam energy in both cases. To do this, the size of the obstacle and its location must be different for different beams. The work can be published only after the authors prove the correctness of comparing the self-healing of two different beams.
Reviewer 4 Report
The authors of the article presented new results of the study of power-exponent-phase vortex beam with nonlinear phase winding.
The text of the article was clearly structured, and allows readers to evaluate the quality of the research presented.
A great advantage was the verification of the numerical results of modeling on the basic optical experiment using SLM.
There were no obvious errors in the text.
The disadvantage of this work was the absence of the "Discussion" section, in which the authors could describe the advantages and disadvantages of the presented method, and further prospects for its use.
Reviewer 5 Report
In this article, the authors discussed the influence of the airy function on the new power-exponent-phase vortex beam. They claim that the Airy transform of the new exponent-power-phase function improves its self-healing property. This study is interesting and could find importance in FSO communication and optical trapping. The manuscript is well written and presented, however, I have the following comments;
-
The introduction section is well written and covered almost every aspect of the vortex beam and the new vortex beam, however, it is lacking in explaining the ways to generate and characterize the vortex beams. In my opinion, for the completeness of the literature review, authors should include a proper introduction to the generation and detection of vortex beams (https://doi.org/10.1515/nanoph-2018-0072, https://doi.org/10.1016/j.optcom.2020.126710, https://doi.org/10.1038/s41377-019-0194-2 ). And compare them with the case of a new power-exponent-phase vortex beam.
-
The authors didn't discuss the parameters ? used in eq. 1. Is it the notations for azimuthal coordinate? If yes then why authors used ? for the azimuthal coordinate in the introduction part?
-
The topological charge of a new power-exponent-phase vortex beam as shown in ref 29, can be easily calculated by either spotting points of minima or intensity lobes. However, in the case of the present manuscript, it does not seem to be that simple (see Fig. 3), as the intensity of the side lobes has reduced. Please comment on that.
The manuscript is interesting and can be accepted for publication after incorporating the comments.
